# Surgery for Port-Wine Stains: A Systematic Review

**DOI:** 10.3390/jpm13071058

**Published:** 2023-06-28

**Authors:** Giacomo Colletti, Sara Negrello, Linda Rozell-Shannon, Gregory M. Levitin, Liliana Colletti, Luigi Chiarini, Alexandre Anesi, Mattia Di Bartolomeo, Arrigo Pellacani, Riccardo Nocini

**Affiliations:** 1The Vascular Birthmark Foundation, P.O. Box 106, Latham, NY 12110, USA; giacomo.colletti@unimore.it (G.C.); vbfpresident@gmail.com (L.R.-S.);; 2Department of Medical and Surgical Sciences for Children & Adults, Cranio-Maxillo-Facial Surgery, University of Modena and Reggio Emilia, Largo del Pozzo 71, 41124 Modena, Italy; 3Cranio-Maxillo-Facial Surgery Unit, University Hospital of Modena, 41124 Modena, Italy; 4Department of Biomedical, Surgical and Dental Sciences, University of Milan, Via della Commenda 10, 20122 Milan, Italy; 5Surgery, Dentistry, Maternity and Infant Department, Unit of Dentistry and Maxillo-Facial Surgery, University of Verona, P.le L.A. Scuro 10, 37134 Verona, Italy; 6Section of Ear Nose and Throat (ENT), Department of Surgical Sciences, Dentistry, Gynecology and Pediatrics, University of Verona, 37124 Verona, Italy

**Keywords:** port-wine stains, capillary malformations, surgery, tissue hypertrophy, laser therapy

## Abstract

Background: Port-wine stains (PWS) are congenital low-flow vascular malformations of the skin. PWS tend to become thicker and darker with time. Laser therapy is the gold standard and the first-line therapy for treating PWS. However, some resistant PWS, or PWS that have tissue hypertrophy, do not respond to this therapy. Our aim is to evaluate the role of surgery in the treatment of PWS birthmarks. Methods: A literature search was performed in PubMed, Scopus, Web of Science (WOS) and Google Scholar for all papers dealing with surgery for port-wine stains, from January 2010 to December 2020 using the search strings: (capillary vascular malformation OR port-wine stains OR Sturge Weber Syndrome OR sws OR pws) AND (surgical OR surgery). Results: Ten articles were identified and used for analysis. They were almost all case series with a short follow up period and lacked an objective–systematic score of evaluation. Conclusions: Delay in treatment of port wine stains may result in soft tissue and bone hypertrophy or nodules with disfiguring or destructive characteristics. The correction of PWS-related facial asymmetry often requires bone surgery followed by soft tissue corrections to achieve a more harmonious, predictable result.

## 1. Introduction

Capillary malformations (CMs), also called port-wine stains (PWS), are one of the most common congenital vascular malformations of the skin. Their incidence is approximately 0.3% to 0.5% of newborns without sex predilection [1]. 

The most frequently affected area is the face, followed by the neck, trunk, and extremities [2]. Most facial PWS follow the cutaneous distribution of the trigeminal nerve, particularly the V2 dermatome. PWS can also involve the oral mucosa, gingiva, larynx, and nose, resulting in gingival bleeding, dysphonia, dysphagia, upper airway obstruction, or epistaxis. 

Unlike Infantile Hemangiomas, which are vascular neoplasms, PWS are present at birth and do not resolve spontaneously but persist throughout life, growing parallel to the child’s growth [3,4]. They often seem to lighten considerably in the first few weeks of life, but this is likely due to a temporary decrease in hematocrit [5]. 

CMs generally appear as flat pink to red macules in toddlers and can be mistaken for a bruise, or they can be overlooked because they are masked by the erythema of neonatal skin [2,6]. 

The PWS tends to become thicker and darker during adult life due to progressive ectasia of the superficial cutaneous vascular plexus. This may be due to a defect in sympathetic and even sensory perivascular innervation with an altered modulation of vessel tone [2,4]. 

Over time, approximately two-thirds of untreated PWS may develop progressive soft tissue and bony hypertrophy and produce nodules prone to spontaneous bleeding. This is particularly true for facial lesions [7,8]. 

According to some authors, this could be explained by an increased vascular permeability, leading to an increased leakage of extracellular nutrients and related growth factors.

The mainstay and first-line therapy for focal PWS is still the pulsed dye laser (PDL) because it allows for good selective vascular obliteration [9,10]. However, this laser is used only for very superficial malformations because it has a penetration depth of less than 2 mm [9,10]. 

According to the literature, PDL laser has low efficacy on longstanding CMs with soft tissue hypertrophy and nodular lesions, with an increased risk of scarring [11,12,13]. 

Moreover, it is difficult to achieve a complete clearance in 20–30% of the lesions even with early laser therapy, and, sometimes, the effects of PDL may reach a plateau after several treatments with redarkening of the lesions secondary to revascularization after years of follow-up [13]. 

Over time, different lasers, such as Nd: YAG lasers, have been used in the treatment of resistant or hypertrophied CMs. However, these lasers have been demonstrated to improve soft tissue hypertrophy, but with a higher frequency of local complications such as scabs, pigmentary alterations, and scars.

To treat hypertrophy and resistant port-wine stains, a possible solution could be surgical correction, despite the potential for considerable morbidity and aesthetic disturbances.

There is little knowledge of the surgical outcomes, and there is a lack of systematic studies on the treatment of disfiguration caused by PWS.

## 2. Materials and Methods

We conducted a systematic review in accordance with the Preferred Reporting Items for Systematic reviews and Meta-Analyses (PRISMA Statement 2009) [14]. 

### 2.1. Search Questions

The search questions were: “What is the role of surgery in the treatment of PWS?” and “What kind of surgery does a PWS need?” 

### 2.2. Search Strategy

We conducted a literature search in PubMed, Scopus, and Web of Science (WOS) for all articles within surgery for port-wine stains published between January 2010 and December 2020 using the search strings: (capillary vascular malformation OR port-wine stains OR Sturge Weber Syndrome OR sws OR pws) AND (surgical OR surgery).

An adjunctive search was conducted in Google Scholar using the same strings.

### 2.3. Inclusion Criteria

According to the PICOS scheme, we adopted the following inclusion criteria [14]: (P) Patients: Patients with capillary vascular malformations regardless of gender or age (I) Intervention: Patients surgically treated for port-wine stains (C) Comparator: not applicable; (O) Outcomes: aesthetic results and incidence of complications and recurrence. (S) Study design: Clinical human studies, including randomized case-control studies, retrospective studies, and case reports. 

### 2.4. Exclusion Criteria

The following exclusion criteria were applied: 

(1) Patients with other vascular malformations; (2) patients with port-wine stains not surgically treated; (3) studies not in English; (4) animal or in vitro studies; (5) studies published in different journals but on the same patients.

### 2.5. Data Extraction

Two authors (D.B.M. and N.S.) extracted from each study the following data: first author and year of publication, study design, number of patients, type of surgical intervention, and follow-up period.

### 2.6. Quality of Included Studies and Bias

We selected only case reports or case series with a limited number of patients and without any control group. 

The authors have only illustrated their surgical procedures and expressed a personal judgment or patient’s appreciation for the aesthetic results without an objective evaluation score or reporting the evaluated parameters. Above all, most studies focused on Asian patients, whose skin may have different healing properties and, because of a different amount of melanin in the skin, they could develop more adverse effects after laser therapy. Moreover, they could have had different cultural conceptions of beauty and motivations for treatment.

For all these reasons, we could only perform a qualitative review.

## 3. Results

Using the search string, we identified a total of 1657 articles on PubMed, Scopus, and the Web of Science. Then, we removed the duplicates, obtaining 1340 articles. Of these, applying the inclusion and exclusion criteria, we excluded 1323 on the basis of the title and 6 for the abstract (Figure 1). 

After reading the full texts, we selected 10 articles for the analysis, while we excluded one because of incomplete data (Table 1).

The search in Google Scholar retrieved 1610 unfiltered articles. All abstracts for the unfiltered articles were reviewed in terms of their relevance to the subject and produced eight relevant articles that were already present in the previous search.

Below, we briefly report the included studies, reporting strengths and limits for each one.

In 2010, Tark et al. conducted a retrospective study on 15 patients who had long-standing PWS on the face. He attempted to radically remove the lesion as much as possible without altering the aesthetic units of the face. The resulting defect was covered with a radial forearm-free flap in 12 patients, and in three patients with a skin graft [15].The Authors surmised the usefulness of free flaps to prevent scar contracture.They also conducted a histological study of the removed nodular lesions, concluding that long-standing PWS can be a high-flow malformation with arterial vessels less susceptible to laser.The strengths of this study were:-A long follow-up.
The limitations of this study are:-Variable site and extension of the birthmarks;-Different surgical reconstruction techniques; -The lack of an objective quantitative score;-Based on 15 patients only.
In 2011, Hu et al. conducted a retrospective study on 10 patients with a CM involving almost the entire cheek, treated by surgical resection [16].The defects were mainly covered with an expanded cervical flap. The surgical technique consisted of two stages: during the first phase, the flap was modeled, and in the second, the PWS was resected and the flap rotated to cover the defect. The authors emphasized above all the matching of the color and texture of the reconstruction as compared to the tissues around the reconstructed area.The strengths of this study were:-Based on a well-specified and comparable PWS extension; -Only one surgical technique, although with additional ones in some cases.The limitations of this study were:-A short follow-up period; -The lack of an objective quantitative score;-Ten patients only.A similar reconstructive technique was presented by Chen et al. [17].They published a retrospective study involving eight patients with CM occupying nearly the entire cheek, treated with expanded cervical flaps performed in two stages. The Authors state that all patients reported a good degree of satisfaction, and they pointed out the superiority in color and texture of the flap as compared to skin grafts or free flaps.The strengths of this study were:-A well specified and comparable PWS extension; -Only one surgical technique. The limitations of this study were:-A 10- to 36-month follow-up period;-The lack of an objective quantitative score;-Based on eight patients only.Kim et al. wrote a retrospective study on 25 patients with facial PWS who were surgically treated. The defects were addressed with various types of reconstructions: 5 primary closures, 7 local flaps, 1 expanded flap, 14 split-thickness skin grafts (STSGs), and 11 full thickness skin grafts (FTSGs) [12].The authors reported only one case of incomplete patient satisfaction due to hyperpigmentation of the grafted skin.They concluded that thick STSG can be a good option to cover large defects as compared to FTSGs for the following reasons: (1) no size limit for donor harvesting; (2) higher success rates of the graft; and (3) one-stage surgery.The limitations of this study were:-Various extensions of the lesion;-Different surgical techniques;-A short follow-up; -Lack of an objective quantitative score;-Only 25 patients.Siewiera et al. reported the case of a patient with a PWS that affected about 45% of his body. They treated the patient with a KTP laser. During the treatment, they also performed a surgical reduction of the lower lip [18]. Results were evaluated based on L*a*b coordinates, as suggested by Rah et al. [19]. This is an objective, quantitative assessment of the clearance of the birthmark. Moreover, they tried to design a questionnaire for patient satisfaction.The strengths of this study were:-The authors tried to create a standardized questionnaire. The limitations of this study were:-Only one patient; -Different operators; -The study mainly focused on laser therapy.Cerrati et al. presented a retrospective study of 160 patients with port-wine stains [11]. They surgically treated 87 hypertrophic head and neck PWS resistant to laser therapy. They used a staged zonal approach, designed on the basis of aesthetic facial units, Langer’s lines, and the facial horizontal thirds. When the lesion involved two adjacent dermatomes or horizontal thirds, the resection was performed with a combined or extended approach. However, when the lesion extended over more than two dermatomes or involved multiple facial subunits, the resection was performed in different stages. Either when it was not possible to hide the incisions in the aesthetic units, an elliptical excision was carried out according to the relaxed skin tension line, or a local rotational or advancement flap was used.The authors surmise that a FTSG or local flap is much more evident than the affected skin, and so they did not always remove all the affected skin but rather lightened the skin with laser. The strength of this study was:-A significant number of patients.The limitation of this study was:-Focused on soft tissue hypertrophy.Yamaguchi et al. reported a medical record review of two patients with Sturge–Weber Syndrome (SWS). Patients were treated in two stages. The first step was based on bone hypertrophy correction with orthognathic surgery and ancillary procedures. In the second step, after six months, soft tissue symmetry was restored. Both patients were satisfied with the surgical results [20].This was the first study describing orthognathic surgery for SWS patients. The authors stressed the importance of a multidisciplinary evaluation before performing orthognathic surgery on these patients.They finally recommend a preoperative Cone Beam CT with 3D simulation of the surgery to reduce operative time and thus blood loss.The strengths of this study were:-The correction of both bone and soft tissue hypertrophy; -Only one surgeon; -The first study describing bone surgery in SWS patients. The limitations of this study were:-Two patients only.Short follow-upDong-Han Lee et al. reported their experience with five patients with extensive CMs of the head and neck. For the reconstruction, four patients underwent a thoracodorsal artery perforator (TDAP) free flap, and one patient received a chimeric TDAP free flap with two skin paddles to separately cover defects of the nose and cheek. One patient needed a skin graft to close the donor site. Due to the large volume of this flap, two patients underwent immediate thinning of the flap during the insetting, while delayed flap debulking was performed in three patients. No recurrences were observed [7].The strengths of this study were:-Only extensive CMs of the face were considered;-Only one surgical reconstructive technique. The limitations of this study were:-The Authors did not use an objective quantitative score;-Five patients only;-Only soft tissues were addressed.Dessy and colleagues presented a technique to address upper lip hypertrophy based on unilateral bikini upper lip reduction and a unilateral bull horn resection technique. This technique allowed for the restoration of lip symmetry while hiding the scars in the lip mucosa and in the passage between different aesthetic units, with good aesthetic results while maintaining lip competence and dynamical function. Results were stable at a 2-year follow up [21].The strengths of this study were:-Only a well-specified kind of deformity; -Only one surgical technique. The limitations of this study were:-Two patients only;-Short follow-up period.Jing Zhou et al. described their approach to two patients with both hard and soft tissue hypertrophy. They did a 3D study to plan bone correction [22].First, they corrected bone deformities with orthognathic surgery and facial bone remodeling. After they improved soft tissue symmetry. Both patients were satisfied with the surgical result.The strength of this study was:-Both bone and soft tissues were addressed.The limitations of this study were:-A short follow-up period; -Lack of an objective quantitative score;-Two patients only.

## 4. Discussion

Soft-tissue and bone hypertrophy in PWS represents a therapeutic challenge because three-dimensional tissue deformities cannot be corrected with vascular-specific lasers, and thus surgery becomes the only option [23]. 

In the past, surgical management of port-wine stains was considered contraindicated because torrential hemorrhage was anticipated. Now, however, we know that a patient with PWS is operable with only a manageable increase in bleeding risk.

The pathological vessels, in fact, are located in the superficial dermis, even though hypertrophy involves all the tissue layers (including the bone). Thus, deep tissues are usually not characterized by marked hypervascularity. For this reason, an increased risk of hemorrhage only exists during the initial skin incision. Moreover, hypothetical bleeding can be easily managed because these are low-flow malformations and consist of small-diameter vessels. 

Another issue is that some of these patients are syndromic, with a greater anesthetic risk, as pointed out by Yamaguchi et al. In their opinion, this risk could be reduced by relying on a multidisciplinary team and accurate preoperative planning [20]. 

Tark et al., Kim et al., and Dong-Han Lee described various soft tissue debulking techniques with an improvement in facial profile and symmetry [7,12,15]. 

According to the authors, the treatment of choice is resection and primary closure; however, this can only be carried out in small lesions. In most cases, it is necessary to cover large soft tissue defects and prevent skin contracture, which can cause eyelid and lip malpositions, nostril deformity, and others.

To restore the resulting defects, Kim et al. performed various types of reconstructions: local flaps, expanded flaps, split-thickness skin grafts (STSGs), and full-thickness skin grafts (FTSGs).

FTSG is a traditional, but not acceptable, method. It often ends in an unaesthetic appearance with skin contractures and facial disfigurement. According to Kim, STSGs are a better choice because the grafts have a better survival rate.

However, all skin grafts have a much different appearance as compared to the tissue to be reconstructed, and a local or adjacent flap seems to be a better choice, having similar color and texture, but major resections may require other, wider flaps.

Free perforator flaps are an option in this case, and they are particularly useful to reconstruct wide and thicker lesions, as reported by Tark and Dong-Han Lee [7,15]. They prevent the risk of contracture and are a single-stage procedure. However, they often lead to poor aesthetic results because of their different color and texture as compared to the surrounding facial skin.

Another trick to solve the aesthetic problem could be the use of expanders that can increase the size of donor tissue without concern for skin contracture and mismatching [16,17]. 

In spite of their potential good aesthetic result, expanded flaps require multiple frequent sessions to achieve the desired size and two surgical stages.

All authors agreed on the need to conceal scars and preserve aesthetic facial units.

Cerrati et al. retrospectively defined the surgical management for each area of the face using a staged zonal approach, designed on the basis of aesthetic facial units and Langer’s lines [11]. 

The middle third of the face is the most commonly involved site, particularly the midline of the upper lip [1,2]. The upper lip, specifically, represents a crucial aesthetic facial unit. When a part of the lip is removed, it should be restored with a tissue as similar in color and texture as possible, trying to reconstruct the normal contour of the lip itself and the physiological transition between the skin and the mucosa [24]. 

Dessy et al. proposed reducing the volume of the hypertrophied upper lip by combining a bikini lip reduction technique with a modified bull horn lip lift. This technique allowed for lip symmetry restoration while hiding the scars in the shadow areas of the face and in the passage between different aesthetic units, with good aesthetic results [21]. 

Even though the authors mentioned the need for bone recontouring in some patients, they did not really address bony overgrowth, which is typical in patients with long-standing PWS. 

There is little awareness of the facial bone hypertrophy caused by PWS and its surgical correction.

Venous malformations of the facial bones are better understood, and they have been divided into two types: the central and the peripheral types. The central one derives from the spongiosa vessels, while the peripheral type arises from periosteal vessels and, secondarily, affects the bone. In these cases, an en-bloc resection is usually performed in order to minimize bleeding, especially in the central type [25,26]. 

In contrast, in PWS, bon hypertrophy appears to result from local hypervascularity around either the periosteum or the spongiosa, or both [27]. Consequently, radical surgery is not a rational option.

Kazuaki Yamaguchi et al. first reported the use of orthognathic surgery in patients with SWS. The authors suggested first addressing skeletal and dental issues and then repositioning the soft tissue to achieve a more aesthetic and symmetric appearance [20]. 

Similarly, Zhou suggested first treating the skeletal deformity and then, after that, addressing the soft tissues. However, in the cases they reported, dentofacial disturbances were minor, and the authors thus focused on correcting the zygomatic and mandibular shapes. Despite having to undergo a multi-step procedure, the patients were willing to face it. This was caused by the fact that they perceived the aesthetic appearance caused by the PWS as highly impacting in terms of quality of life [22]. 

Our paper has some limitations: first, the small number of articles on the surgical treatment of PWS and the small sample size of each. Also, the morphological issues that were addressed in the single papers were highly different and thus difficult to compare. 

All the publications were retrospective without any prospective studies.

Many of these studies were penalized by a short follow-up period, and this does not allow for verification of the stability of the surgical results.

There was a significant prevalence of papers dealing with Asian patients. This may carry the risk of bias due to peculiar physiognomic features and a particular cultural background.

Finally, no author used standardized and reproducible evaluation protocols but rather relied on personal judgment.

For all these reasons, it was not possible to draw significant conclusions from our review, but only qualitative observations.

## 5. Conclusions

This systematic review could not lead us to infer anything in terms of indications or preferable surgical procedures.

A higher number of cases have to be analyzed, and a longer-term follow-up is highly desirable. Hypertrophy in PWS has a tendency towards relapse, and the results may accordingly not be stable over time. 

However, we can draw some useful conclusions. Minor hypertrophic changes could be treated by simple camouflage procedures, while major disease-related disfigurement often requires major, multi-step surgeries. Pre-expanded flaps seem to offer the best results when large areas have to be corrected, especially in the cheek. Bony deformities, which are very often present in PWS, should be treated as the first surgery in order to maximize the final result. 

In these patients, bleeding and anesthetic risks could be addressed by a multidisciplinary team with accurate preoperative planning.

## Figures and Tables

**Figure 1 jpm-13-01058-f001:**
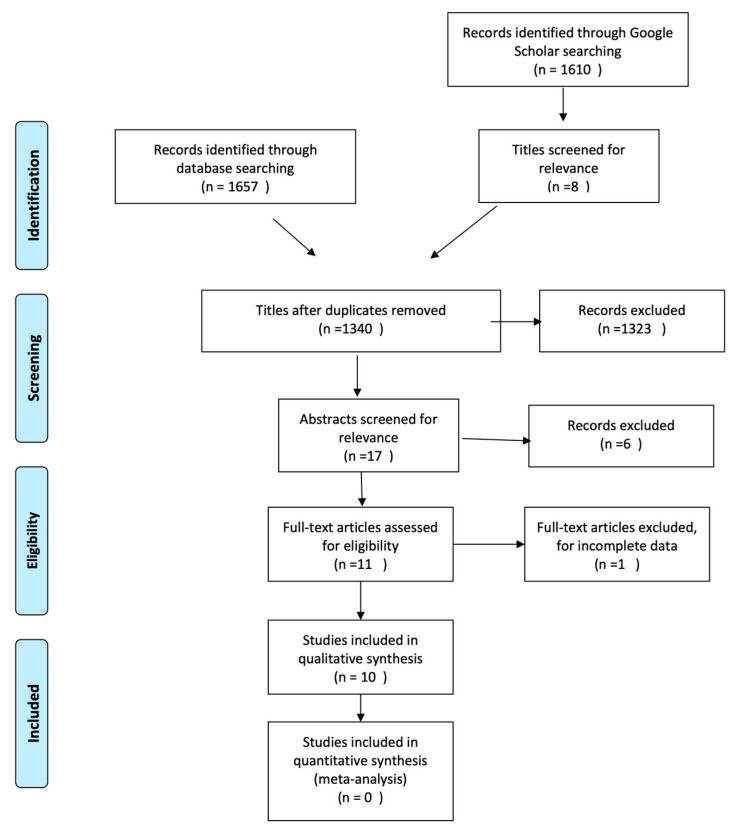
Search strategy according to PRISMA guidelines.

**Table 1 jpm-13-01058-t001:** Summary of study characteristics and results.

Authors	Year	Title	Patients	Surgery	Results	Relapse	Follow-Up
**Tark et al.**	2010	The Fate of Long-Standing Port-Wine Stain and Its Surgical Management	15	3 skin grafts 12 radial forearm free flap	Good satisfaction	no	12 years
**Hu et al.**	2011	Reconstruction of the Cheek after Large Port-Wine Stain Lesion Resection	10 Large cheek PWSs	Prefabricated expanded cervical flaps carried by the superficial temporal vessels	2 failures 8 good results	no	2–22-months
**Chen et al.**	2019	Surgical Treatment for Facial Port-Wine Stain by Prefabricated Expanded Cervical Flap Carried by Superficial Temporal Artery	9 Large cheek PWs	Prefabricated expanded cervical flaps carried by the superficial temporal vessels	1 failure 8 good results	no	10–36 months
**Kim et al.**	2012	Surgical Treatment of Dermatomal Capillary Malformations in the Adult Face	25 various facial and neck PWSs	5 primary closures, 7 local flaps, 1 expanded flaps, 14 STSGs, 11 FTSGs.	1 STSG moderately satisfied 24 good satisfaction	no	13.2 months
**Siewiera et al.**	2012	Combined Laser and Surgical Treatment of Giant Port-Wine Stain Malformation—Case Report	1 PWS covering about 45% of body area	55 KTP laser procedures and a lower lip reduction	Good satisfaction	no	1 year
**Cerrati et al.**	2014	Surgical Treatment of Head and Neck Port-Wine Stains by Means of a Staged Zonal Approach	160 various PWSs	surgical treatment based on a subunit and zonal approach to the face	Good satisfaction	no	
**Yamaguchi et al.**	2016	Correction of Facial Deformity in Sturge–Weber Syndrome	2 SWS	Orthognathic surgery and facial bone contouring and then reduction and reposition of soft tissue	Good satisfaction	no	49.4 months 14 months
**Dong-Han Lee**	2016	Reconstruction of Head and Neck Capillary Malformations With Free Perforator Flaps for Aesthetic Purposes	5 extensive CMs of the head and neck	1 chimeric TDAP free flap 4 TDAP free flaps	1 flap underwent went thrombectomy and reanastomosis due to arterial insufficiency, 1 flap marginal partial necrosis Good long term results	no	Mean FUP 35 months
**Dessy et al.**	2018	Surgical correction of hypertrophic upper lip in vascular malformations	2 patients with upper lip hypertrophy due to CMs 3 patients with upper lip hypertrophy due to LMs	asymmetric bikini upper lip reduction and unilateral bull horn resection technique	Good satisfaction	no	2 years
**Jing Zhou et al.**	2020	Surgical correction for patients with port-wine stains and facial asymmetry	2 facial CMs	first bone and dental correction and then a reposition of the soft tissue	Good satisfaction	no	4 years 2.5 years

## Data Availability

Not applicable.

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
