# Peer review of "Surgery for Port-Wine Stains: A Systematic Review"

_jpm, 2023, doi:10.3390/jpm13071058_

Round 1
Reviewer 1 Report
I applaud the authors for putting together a lot of information in this article, which is a review of surgical treatment of port wine stains. There is some good information here, however in the end, only 10 articles were identified and used for the analysis. One of the issues is the only included patients who had some sort of surgical treatment. I realize that is the title of the article, but surgical interventions, as the article acknowledges, are challenging, and oftentimes the results are not cosmetically appealing. The authors say on page 2, that the only effective solution is surgical correction for resistant and hypertrophic port wine stains, is not entirely correct, as deeper penetrating lasers do have some effect on deeper port wine stains.
The lack of photographs included with some of these retrospective articles is also disappointing, as this is a retrospective review of multiple retrospective reviews.
Certainly for deeper and more robust port wine stains deeper penetrating lasers have been used, sometimes with good results and unfortunately, sometimes with side effects. However after reading this article multiple times the story that it weaves is one that is not that helpful for the reader as far as managing port wine stains, which is really what one would hope in an article like
will the article help in making good decisions based on the literature. The way this article flows and the types of articles that were used, offers little aid for either counseling of the patient or treating a patient.
Author Response
Dear Reviewer,
we would like to thank you for the useful suggestions, which allowed us to improve our paper. We have now prepared a revised version of the manuscript, where all the changes have been introduced. In particular, the questions raised in your report will be tackled one by one in what follows.
We hope that, with these corrections and additions, you will now be able to accept our paper.
Best regards,
Sara Negrello
Comments and Suggestions for Authors
I applaud the authors for putting together a lot of information in this article, which is a review of surgical treatment of port wine stains. There is some good information here, however in the end, only 10 articles were identified and used for the analysis. One of the issues is the only included patients who had some sort of surgical treatment. I realize that is the title of the article, but surgical interventions, as the article acknowledges, are challenging, and oftentimes the results are not cosmetically appealing.
Answer
This study analyzes only 10 articles because only 10 articles reflect the characteristics sought. The authors, in fact, wanted to evaluate the role of surgery in the treatment of capillary malformations and not to analyze the role of the various therapeutic tools in the treatment of port wine stains.
The authors say on page 2, that the only effective solution is surgical correction for resistant and hypertrophic port wine stains, is not entirely correct, as deeper penetrating lasers do have some effect on deeper port wine stains. Certainly for deeper and more robust port wine stains deeper penetrating lasers have been used, sometimes with good results and unfortunately, sometimes with side effects.
Answer
Yes, there are deeper lasers that can be used to treat hypertrophic port-wine stains, however they often come with a higher risk of scarring. We believe it is a more acceptable cosmetic result to have surgical scars hidden in wrinkles or gray areas. Also, the laser cannot be used in bone hypertrophy. As mentioned, obviously, both surgical and laser treatment must be carried out in expert hands.
The lack of photographs included with some of these retrospective articles is also disappointing, as this is a retrospective review of multiple retrospective reviews.
However after reading this article multiple times the story that it weaves is one that is not that helpful for the reader as far as managing port wine stains, which is really what one would hope in an article like will the article help in making good decisions based on the literature. The way this article flows and the types of articles that were used, offers little aid for either counseling of the patient or treating a patient.
Answer
As we said in the conclusions, the low number of patients evaluated does not allow real guidelines, however it does allow for some observations that we deem useful. First, of course, the fact that it is desirable to treat the hypertrophy of the bone bases, and only then the hypertrophy of soft tissues. Second, the use of expanders can be a good starting point for other surgeons who have difficulty choosing reconstruction in such a critical area as the face.
Reviewer 2 Report
Congratulations for taking the time of compiling such a big amount of practical information regarding this poorly studied therapeutical option in capillary malformations
Author Response
Dear Reviewer,
we would like to thank you for the useful suggestions, which allowed us to improve our paper. We have now prepared a revised version of the manuscript, where all the changes have been introduced. In particular, the questions raised in your report will be tackled one by one in what follows.
We hope that, with these corrections and additions, you will now be able to accept our paper.
Best regards,
Sara Negrello
Quality of English Language
I am not qualified to assess the quality of English in this paper
Is the work a significant contribution to the field? 4 |
|
Is the work well organized and comprehensively described? 4 |
|
Is the work scientifically sound and not misleading? 4 |
|
Are there appropriate and adequate references to related and previous work? 5 |
|
Is the English used correct and readable? 4 |
Comments and Suggestions for Authors
Congratulations for taking the time of compiling such a big amount of practical information regarding this poorly studied therapeutical option in capillary malformations
Answer
Thanks for your kind review and your time
Reviewer 3 Report
Table 1 should be rewritten, as the selected line spacing makes it difficult to use.
On line 178 instead of pws should be PWS.
Probably the references should be rewritten, according to the journal's requirements, and included in the text using square brackets, instead of round ones.
The English language is fine.
Author Response
Dear Reviewer,
we would like to thank you for the useful suggestions, which allowed us to improve our paper. We have now prepared a revised version of the manuscript, where all the changes have been introduced. In particular, the questions raised in your report will be tackled one by one in what follows.
We hope that, with these corrections and additions, you will now be able to accept our paper.
Best regards,
Sara Negrello
Quality of English Language
(x) Minor editing of English language required
Is the work a significant contribution to the field? 1 |
|
Is the work well organized and comprehensively described? 4 |
|
Is the work scientifically sound and not misleading? 3 |
|
Are there appropriate and adequate references to related and previous work? 3 |
|
Is the English used correct and readable? 4 |
Comments and Suggestions for Authors
Table 1 should be rewritten, as the selected line spacing makes it difficult to use.
Answer:
Done, thank you.
On line 178 instead of pws should be PWS.
Answer:
Done, thank you.
Probably the references should be rewritten, according to the journal's requirements, and included in the text using square brackets, instead of round ones.
Answer:
Done, thank you.
Comments on the Quality of English Language: The English language is fine.
Round 2
Reviewer 1 Report
paper is better, just need to include that there are laser options beyond PDL for resistant PWS
Author Response
Dear Reviewer,
we would like to thank you for the useful suggestions, which allowed us to improve our paper. We have now prepared a revised version of the manuscript, where all the changes have been introduced. We hope that, with these corrections and additions, you will now be able to accept our paper.
Best regards,
Sara Negrello
Comments and Suggestions for Authors
Paper is better, just need to include that there are laser options beyond PDL for resistant PWS
Answer
On page 2 we have specified the presence of deeper lasers for the treatment of hypertrophic or resistant PWS. We also specified that we suggest surgery as a good therapeutic option and not as the only solution.